# Population-based Evaluation in Repeated Rock-Paper-Scissors as a Benchmark for Multiagent Reinforcement Learning

**Marc Lanctot**                                                    *lanctot@google.com*
*Google DeepMind*

**John Schultz**                                                    *jhtschultz@google.com*
*Google DeepMind*

**Neil Burch**                                                      *nburch@ualberta.ca*
*Sony AI*
*University of Alberta*

**Max Olan Smith**                                                  *mxsmith@umich.edu*
*University of Michigan*

**Daniel Hennes**                                                   *hennes@google.com*
*Google DeepMind*

**Thomas Anthony**                                                  *twa@google.com*
*Google DeepMind*

**Julien Pérolat**                                                  *perolat@google.com*
*Google DeepMind*

**Reviewed on OpenReview:** *https://openreview.net/forum?id=gQnJ7ODIAx*

## Abstract

Progress in fields of machine learning and adversarial planning has benefited significantly from benchmark domains, from checkers and the classic UCI data sets to Go and Diplomacy. In sequential decision-making, agent evaluation has largely been restricted to few interactions against experts, with the aim to reach some desired level of performance (e.g. beating a human professional player). We propose a benchmark for multiagent learning based on repeated play of the simple game Rock, Paper, Scissors along with a population of forty-three tournament entries, some of which are intentionally sub-optimal. We describe metrics to measure the quality of agents based both on average returns and exploitability. We then show that several RL, online learning, and language model approaches can learn good counter-strategies and generalize well, but ultimately lose to the top-performing bots, creating an opportunity for research in multiagent learning.

## 1 Introduction

How should agents be evaluated when learning with other learning agents? One metric is simply the average return over an agent's lifetime. Another is the agent's robustness against a potential nemesis whose goals are only to minimize the agent's return. The first is the conventional metric used in the evaluation of reinforcement learning (RL) agents, while the second is quite common among game-theoretic AI techniques for imperfect information games. In this paper, we argue our position that neither of these is generally sufficient in isolation: good agents should both maximize return *and* be robust to adversarial attacks.

|          |     | R | P | S |
|----------|-----|---|---|---|
|          |     | $R$ | $P$ | $S$ |
|          | $R$ | $(0, 0)$ | $(1, -1)$ | $(-1, 1)$ |
| Player 0 | $P$ | $(1, -1)$ | $(0, 0)$ | $(-1, 1)$ |
|          | $S$ | $(-1, 1)$ | $(1, -1)$ | $(0, 0)$ |

Player 1

Figure 1: Rock, Paper, Scissors. Player 0 chooses an action assigned to a row, and similarly player 1 for a column. Each entry shows the reward for player 0, then player 1 respectively.

The classical method to demonstrate superior AI performance is head-to-head matches, or direct comparisons of average return, against the strongest known agents. This method has driven progress of the field since the beginning: from Samuel's checkers program, to chess, Go, poker, modern real-time games, and so on. On the other hand, game-theoretic approaches to learning result in agents that approximately respond to a population of opponents which are enumerated in hopes that the full strategic complexity of the game is captured among the set of opponents, and convergence to an approximate Nash equilibrium is obtained. The extent to which current AI systems are robust to adversarial attacks is unclear. Nevertheless, there is evidence that even expert level AI agents can be demonstrably susceptible to adversarial behavior (Timbers et al., 2022; Wang et al., 2022). While current evaluation methodologies over-emphasize the single metric of cumulative reward or performance against experts, human or AI, we argue that the more important problem is the lack of benchmarks that prioritize the evaluation of agents in a more general way, where multiple metrics could lead to a better understanding of an agent's capabilities.

In this paper, we propose a benchmark based on the classical game of Rock, Paper, Scissors augmented in two ways: first, it is a repeated game and hence a sequential decision-making problem; second, performance is measured against a population of agents with varied skill. The simplicity of the stage game is of paramount importance: it is a well-understood two-player zero-sum game whose game-theoretic optimal strategy is well-known, and by construction maximizing rewards against fallible opponents naturally leads to behavior that is potentially exploitable. For learning agents to find exploits in the opponents, they must correctly deduce their strategies from observations. We describe a population of forty-three openly-available hand-crafted agents that were submitted to competitions and characterize their head-to-head performance, exploitability, and the extent to which they are predictable (by supervised learning). We then train agents using several modern approaches with different capabilities, against the population and independently trained against copies of themselves. These approaches show promise in various ways: out-of-distribution generalization of exploitative behavior, a clear lack of exploitable behavior, and a good balance between these two metrics. Ultimately, none of the agents are able to outperform the top two participants in head-to-head matches while being more robust to exploits, leading to a challenge and opportunity for novel multiagent reinforcement learning research.

## 2 Repeated Rock, Paper, Scissors

In this section, we describe the basic notations, the environment, competition and participants, and population-based evaluation. The environment and population are freely available within OpenSpiel (Lanctot et al., 2019).

### 2.1 Notation and Environment Description

A normal-form game has a discrete set of players $\mathcal{N} = \{1, 2, \cdots, n\}$. A matrix game is a two-player game with a set of actions per player $\mathcal{A}_1$ and $\mathcal{A}_2$, a joint action set $\mathcal{A} = \mathcal{A}_1 \times \mathcal{A}_2$, and utility functions for each player $i \in \mathcal{N}$, $u_i : \mathcal{A} \to \Re$. A zero-sum game is one where $\forall a \in \mathcal{A}, \sum_{i=1}^{n} u_i(a)$.

Rock, Paper, Scissors (RPS), also called RoShamBo, is a two-player zero-sum matrix game described by the matrix depicted in Figure 1: Rock (R) beats Scissors (S), Paper (P) beats Rock (R), and Scissors (S) beats Paper (P).

The sequential version is repeated: there are $K$ identical plays of RPS. At state $s_0$, agents simultaneously decide their actions and agent $i$ receives intermediate reward $r_{t,i}$ by joint action $a_t$ composed of all agents' actions combined and payoff matrix in Figure 1. A trajectory is a state and (joint) action sequence of experience: $\rho = (s_0, a_0, s_1, a_1, \cdots, s_{K-1}, a_{K-1}, s_K)$. In this environment, every episode has length $K$ and the full (undiscounted) return is defined as $G_{0,i} = \sum_{t=0}^{K-1} r_{t,i}$. We choose $K = 1000$ as a default from the competitions described in Section 2.2.

Similarly to previous work in this environment (Hernandez et al., 2019), observations to the agent depend on the *recall*, $R$. With a $R = 1$, the observation at $s_t$ includes the most recently executed joint action $a_{t-1}$, encoded as a 6-bit observation (two one-hot actions). With $R = 2$, the observation includes the two most recent join actions, and so on, where $R = K$ includes the full action sequence. For example, when $R = 10$ there are $9^{10} \approx 3.5$ billion unique observations; a tabular Q-learning agent would a table of 10.5 billion entries. Unless otherwise noted, use $R = 1$ as a default value.

Finally, as is standard (Sutton & Barto, 2017), a policy $\pi_i$ is a mapping from an observation to a distribution over actions used by agent $i$, and $\pi$ (without subscripts) is the joint policy used by both agents. In RPS, there is a large incentive to use stochastic policies because any deterministic policy is fully exploitable (Shoham & Leyton-Brown, 2009). For simplicity of notation, we denote $G_{t,i,\pi} = \mathbb{E}_{a \sim \pi}[G_{t,i}]$.

## 2.2 Competition and Participants (Bots)

In early 2000s, Darse Billings ran two Repeated Rock, Paper, Scissors (RRPS) competitions (Billings, 2000a;b). In this subsection, we describe the participant entries that were released and still openly accessible, which have since been integrated into OpenSpiel (Lanctot et al., 2019). In each competition, participants were asked to submit a bot[1] to play RRPS, with $K = 1000$, all played within a one-second time limit. Each program had full recall, the entire action sequence in each episode, but nothing more that would identify the other bots. Participants were told in advance that the population would include some sub-optimal bots.

The majority of the entries in the competition were hand-crafted heuristic bots that were developed independently by different programmers. A few participants submitted two entries. The resulting population consists of 43 bots: 25 entrant bots and 18 seed bots from the first competition. Including the winner of the second competition Andrzej Nagorko's GREENBERG, made open-source seperately, and the first competition winner Dan Egnor's IOCAINEPOWDER.

We now summarize the approach taken by most bots. The simplest seed bots do not use their observation to inform their action. RANDBOT generates an action uniformly at random. ROCKBOT always plays rock. R226BOT plays 20% rock, 20% paper, 60% scissors. ROTATEBOT rotates between R, P, S in that order. PIBOT, DEBRUIJN81, TEXTBOT all play a fixed sequence of actions derived from the digits of pi, De Bruijn sequences, and the competition rules in base 3, respectively.

Other seed bots have a recall $R = 1$, i.e. they use only the current observation. SWITCHBOT never repeats its' previous action, and chooses uniformly between the two alternatives. SWITCHALOT repeats previous action with 12% probability; otherwise, chooses uniformly between the two alternatives. COPYBOT plays to beat the opponent's previous action. DRIFTBOT and ADDDRIFTBOT2 bias their action by the opponent's action or joint-action, respectively, with an increase, or "drift", in bias over time. FOXTROTBOT alternates between playing randomly, and an offset of its' previous action.

The remaining seed bots used historical observations either directly or through statistical summaries. FLATBOT3 plays a flat distribution. ADDSHIFTBOT3 biases decision by previous joint action, shifting the bias if losing. ANTIFLATBOT maximally exploits FLATBOT3. ANTIROTNBOT exploits rotations played by the opponent. FREQBOT2 plays to beat opponent's most frequent choice.

The entrant's bots also used historical observations. ROBERTOT uses a voting algorithm informed by observation counts. PREDBOT, PIEDRA, and SWEETROCK predict play from action counts. MOD1BOT models the opponent as PREDBOT. BIOPIC maintains four prediction models differing in available information.

---

[1]In this paper, "bot" always refers to a previous competition participant, whereas "agent" refers to an RRPS player more generally.

MARKOV5, MARKOVBAILS, RUSSROCKER4, and HALBOT inform their prediction with Markov chain models. PHASENBOTT, PETERBOT, MULTIBOT, and MIXED_STRATEGY all switch between a fixed set of policies depending on which is currently the most profitable. INOCENCIO, ZQ_MOVE, MARBLE, GRANITE, BOOM, and SHOFAR also implement complex rule-based decisions informed by summary statistics of the history.

Several bots took very innovative approaches. SUNNERVEBOT implemented a "nervous" network reminiscent of a deep neural network. ACTR_LAG2_DECAY implemented the cognitive architecture ACT-R (Anderson, 1993).

IOCAINEBOT (Egnor, 2000), which won the first competition, works by maintaining a set of predictions about its opponent, and building a set of strategies from each predictor. Predictions included random guessing, frequency analysis, and history matching across six different history sizes. From each prediction six strategies are constructed based on recursive response computations (e.g., triple-guessing). IOCAINEBOT then plays the most historically successful strategy. GREENBERG, by Andrzej Nagorko, won the second competition by extending IOCAINEBOT to include additional predictors utilizing more advanced history matching algorithms.

## 2.3 Population-Based Evaluation

We propose several ways to use this population to evaluate agents. We define an agent's POPULATIONRETURN to be the average return per episode against a bot drawn uniformly at random at the start of the episode. Performance against specific bots can also be reported; we compute the cross-table between all bots in Figure 2. The exploitability of an agent $i$ is by how much their nemesis (best response) beats them. Let $-i$ refer to agent $i$'s opponent. Then,

$$\text{EXPL}(\pi_i) = G_{0,-i,(\pi_i, b(\pi_i))}, \text{ where } b(\pi_i) \in BR(\pi_i), \text{and}$$

$BR(\pi_i) = \{\pi_{-i} | G_{0,-i,(\pi_i, \pi_{-i})} = \max_{\pi'_{-i}} \{G_{0,-i,(\pi_i, \pi'_{-i})}\}\}$ is the set of best responses to $\pi_i$. Notice that exploitability is expressed in the opponent's return; it is non-negative and its lowest value is zero when an agent is not exploitable. However, due to the maximization over the entire policy space, it can be too computationally expensive to compute exactly, so we can approximate it by running several learning algorithms and taking the maximum achievable value. Another measure of approximate exploitability uses the bots as exploiters, taking the maximum over the bots, where $P$ is the population:

$$\text{WITHINPOPEXPL}(\pi_i) = \max_{\pi_{-i} \in P} \mathbb{E}_{a \sim (\pi_i, \pi_{-i})}[G_{0,-i}].$$

Head-to-head performance of all bots in the population is visualized in Figure 2[2]. Each cell represents an average over 1000 episodes. Figure 3 summarizes some properties of the population. First, the population returns of each bot range from $-648.42$ to $288.15$, achieved by GREENBERG. GREENBERG dominates (achieves higher value against all opponents) five bots, and IOCAINEBOT dominates one bot. Second, the within-population exploitabilities range from 1.2 (RANDBOT) to 1000, with several reaching this upper-bound, 316.1 on average. We then trained several RL algorithms until empirical converges (millions of episodes) against each bot independently: Q-learning and IMPALA (Espeholt et al., 2018) with $R \in \{1, 3, 5, 10\}$, and defined the external-learned exploitability of that bot as the maximum value achieved among these eight. These values range from 4.8 to 1000.0, with an average of 420.3. The within-population exploitability achieves 75.2% of the external-learned exploitability on average, and varies between 50-100% of the external-learned exploitability on most bots. Due to this consistency across bots and significantly less computation requirements, we mainly use within-population exploitability from here on.

One simple way to rank agents under both metrics is to assume they both matter equally: AGGREGATESCORE$(\pi_i) =$ POPULATIONRETURN$(\pi_i) -$ WITHINPOPEXPL$(\pi_i)$. This definition makes sense since the units are identical; it could naturally be extended with different weights depending on the evaluation context. These two metrics (population return and within-population exploitability) capture the essence of the RPS game in the repeated setting: the only way another agent can predict an agent's next action is

---

[2]The precise values in this table are available from OpenSpiel (Lanctot et al., 2019): `https://github.com/google-deepmind/open_spiel/tree/master/open_spiel/data/paper_data/pbe_rrps`

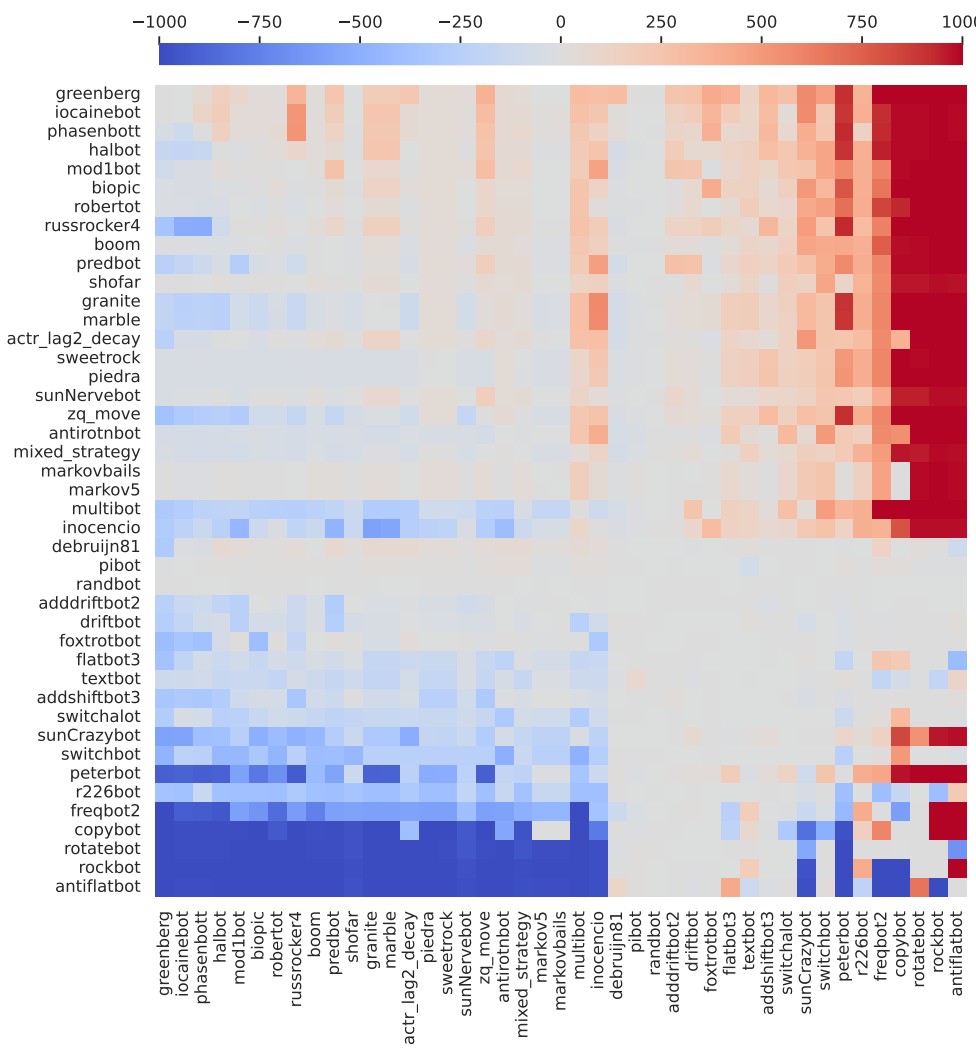

Figure 2: RoShamBo bots payoff table. Each cell shows the return for the row bot versus the column bot averaged across 1000 episodes.

by determining their learning rule from the history of choices made by both agents. Ultimately the goal is to maximize return, but finding a best response to the other player's action choice is the entire game, so agent cannot be too exploitable in doing so without risking giving up reward to its opponents from the population. Hence, the combined score acts as a summary of how well an agent is performing on both fronts within is population, allowing agent designers to compare a single number. Under this metric, we list the top 10 bots in Table 1. The complete ranked list is given in Appendix A.1. For reference, we also include the scores of the best learning algorithm in each category from Secton 4.

## 3 Predictability of RPS Bots

In order to win at RRPS, the bots must attempt to predict the actions chosen by other bots, while not becoming predictable themselves by their opponent. In this section, we investigate to what extent the bots are predictable by a neural network.

To assess how predictable each bot was, we sampled games of RRPS between the bot and each other bot, including itself. We trained an LSTM per bot to predict that bot's next action with recall $R = 20$ (details

| Bot Names | Pop. Return | W.P. Expl. | Agg. Score |
|---|---|---|---|
| GREENBERG | 288.15 | 3.65 | 284.50 |
| IOCAINEBOT | 255.00 | 5.00 | 250.00 |
| BIOPIC | 196.36 | 36.66 | 159.70 |
| BOOM | 169.11 | 27.93 | 141.19 |
| SHOFAR | 152.01 | 16.87 | 135.14 |
| ROBERTOT | 177.77 | 50.16 | 127.61 |
| PHASENBOTT | 232.25 | 111.71 | 120.54 |
| MOD1BOT | 203.16 | 90.16 | 113.00 |
| SWEETROCK | 146.25 | 41.21 | 105.04 |
| PIEDRA | 146.08 | 41.44 | 104.64 |

| Algorithm/Agent Names | Pop. Return | W.P. Expl. | Agg. Score |
|---|---|---|---|
| PopRL | 258.00 | 10.98 | 247.02 |
| LLM (Chinchilla 70B) | 201.00 | 45.80 | 155.20 |
| ContRM | 164.77 | 16.27 | 148.51 |
| QL ($R = 10$) | $-0.52$ | 8.62 | 8.10 |
| R-NaD | $[-10, 5]$ | $[20, 40]$ | $[-50, -25]$ |

Table 1: Top 10 bots ranked by AGGREGATESCORE, and top learning algorithms in each category from subsections of Section 4. The bot results are computed on 1000 episodes per profile. The algorithm results are averaged over 5 seeds.

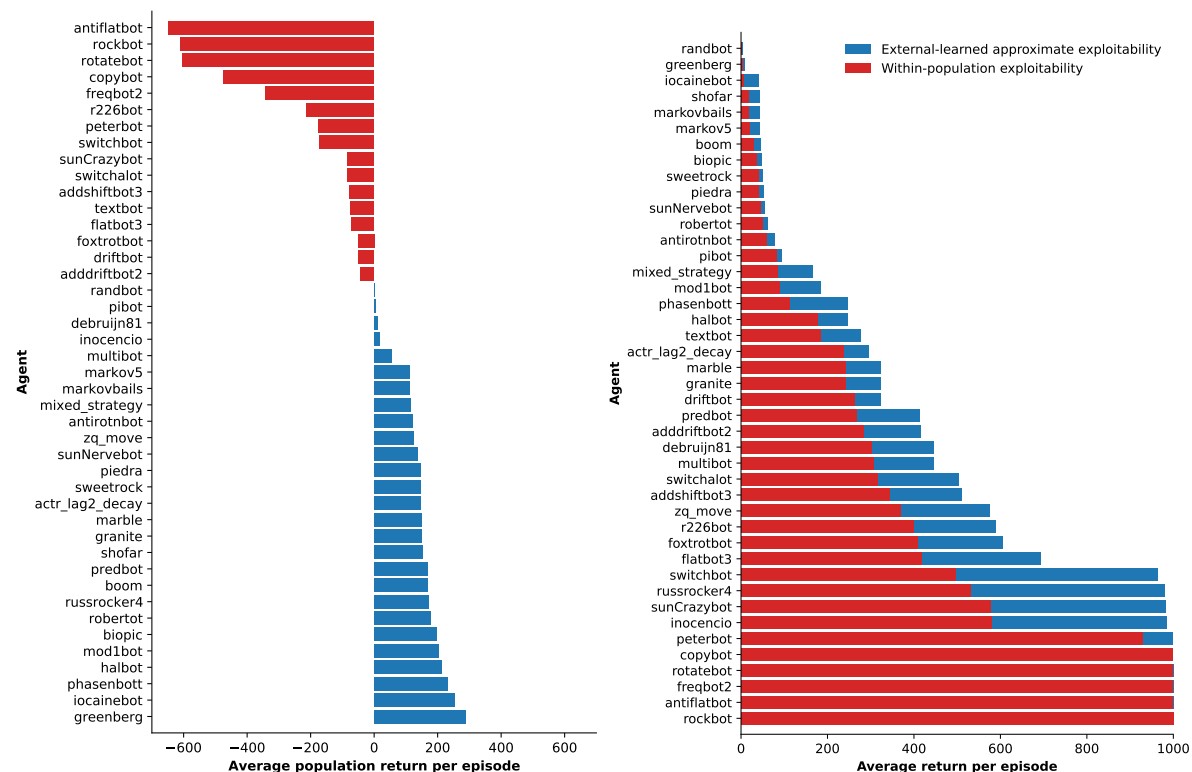

Figure 3: Left: Population Returns for each bot. Right: Approximate exploitabilities for each bots.

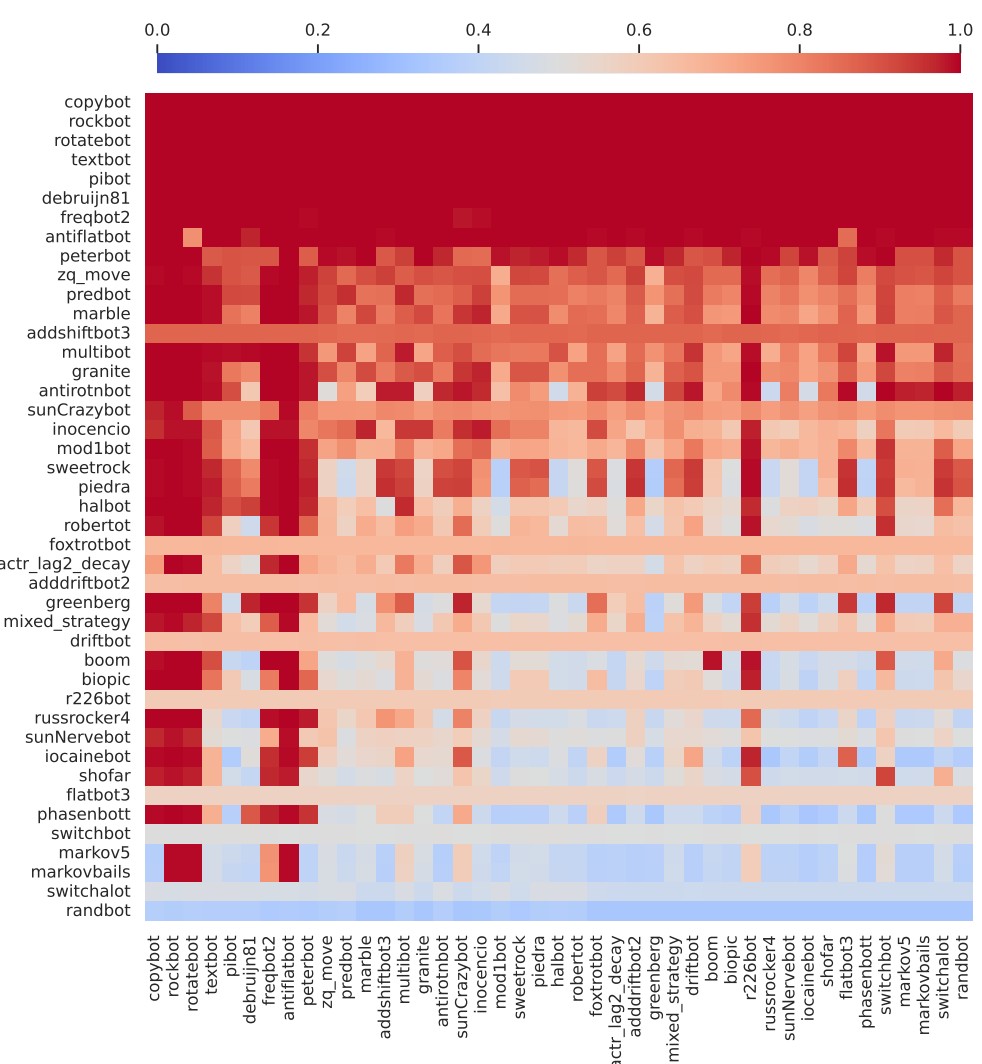

Figure 4: Action prediction accuracy. Each cell shows the action prediction accuracy for row bot versus the column bot averaged across 100 episodes.

in App. A.5). We report the prediction accuracy, i.e. the proportion of the time that the predicted action matched the bot's action, shown in Fig. 4.

Some bots are deterministic and easy to predict, e.g. ROCKBOT was predicted correctly 100% of the time. Stochastic bots, such as RANDBOT, have low predictability, but this comes at the cost of their ability to exploit other bots. Prediction accuracy for the entrants was substantially greater than for the Nash equilibrium, but varied substantially from 48% for MARKOVBAILS to 94% for PETERBOT.

Successful action prediction reveals the existence of structure within the bot population. In principle, RRPS is a purely non-transitive game, and there is no such thing as a 'better' strategy. Under a unique Nash equilibrium, an agent's past actions are not predictive of their future actions. Still, we hypothesize that it is possible to learn action predictions from a sub-population that generalise to the whole population.

To test this, we sample 30 bots from the population randomly, and generate RPS games between these bots. We then train the same LSTM model as before to predict the bots' actions. To succeed at this task, the agent must identify the strategy a bot is employing to predict the next action as it no longer knows the bot identities a priori. This also means that if held-out bots employ similar strategies, the agent should be able

|  |  | co-player | |
|---|---|---|---|
|  |  | train | test |
| predicted bot | train | 69.14% | 67.35% |
|  | test | 57.80% | 55.65% |

Table 2: Action prediction accuracy.

to predict their actions too. We repeated the experiment 10 times with different splits of training/testing bots.

On games between the training bots, the neural network achieved an average accuracy of 69.1%. In games between the held-out test population, the neural network achieved an accuracy of 55.7%, which is significantly better than chance, and demonstrates there is learnable structure in the bot behaviours. In Table 2, we break down accuracy by whether the bot being predicted or the co-player are in the training population. We show that prediction accuracy drops for either bot or co-player being from the held-out population, but the effect is larger when the bot being predicted is not in the training set.

## 4 Learning to Play Repeated RPS

Can an agent *learn* to earn high population return and not be very exploitable? Here, we show baselines and RL agent performance on this environment. We evaluate them using the population-based evaluation (PBE) criteria in Section 2.3. The resulting best achievable score of each learning approach is summarized in Table 1.

### 4.1 Baseline Independent RL Results

In this section we report the performance of fixed policies and baseline RL agents. Each individual run reports the best achieved performance of an fixed agent or one trained by playing against another copy of an agent of the same type (independent RL). Note that we differentiate this training from "self-play" due to the agents using the same algorithm but separate networks. We then evaluate the agents against the population after 700k - 1M episodes of training, with the population return and returns of each bot against the agent being averaged over a sliding window of the 50 most recent evaluations. Each reported value represent the the average over five individual runs using different seeds Table 3.

For Q-learning (QL), we report results over varying recall size $R \in \{1, 3, 5, 10\}$. Interestingly, the within-population exploitability of uniform is greater than zero, which is possible to due to maximization over noisy estimates and the deterministic nature of the random number generators. In addition, we run DQN (Mnih et al., 2015), A2C (Mnih et al., 2016), and Boltzmann DQN (BDQN) (Cui & Koeppl, 2021) with various temperatures $\eta \in \{0.1, 0.5, 1, 2\}$. Overall we found that the algorithms improve as $R$ increases, achieving a population return of at most 18, and can be particularly exploitable. The high exploitability is somewhat mitigated by a sufficiently large ($R = 10$) table in Q-learning, higher temperature in Boltzman DQN, and entropy bonuses in A2C. The best achievable aggregate score across these baselines is 8.1. Appendix A.2 contains hyperparameter selection details for the RL agents.

We then assessed the quality of independent RL agents when the recall is set to a much larger value ($R \in \{100, 1000\}$). This leads to observations that are 10-100 times larger and too many states to fit in memory, so we use only DQN, BDQN, and A2C. We run independent RL for the same amount of wall-clock time as the shorter recall results (12 days), which leads to fewer episodes due to the added computational cost per episode. In all cases, we noticed variation among hyper-parameters, but that many dropped to a low aggregate score (-1600) and then never recovered a higher score. The best values over all the runs are presented in Table 4. The highest achieved aggregate score among these runs was 7.60.

| Name | Pop. Return | W.P. Expl. | Agg. Score |
|------|------------:|-----------:|-----------:|
| ROCK | −610.20 | 1000.00 | −1610.20 |
| PAPER | −613.50 | 999.20 | −1612.70 |
| SCISSORS | −648.10 | 1000.00 | −1648.10 |
| UNIFORM | 0.00 | 9.31 | −9.31 |
| QL ($R = 1$) | −531.28 | 994.54 | −1525.82 |
| QL ($R = 3$) | −280.65 | 910.56 | −1191.21 |
| QL ($R = 5$) | −89.67 | 405.89 | −495.56 |
| QL ($R = 10$) | −0.52 | 8.62 | 8.10 |
| DQN | −194.49 | 693.13 | −887.62 |
| BDQN ($\eta = 0.1$) | −124.52 | 515.60 | −640.12 |
| BDQN ($\eta = 0.5$) | −19.59 | 164.25 | −183.84 |
| BDQN ($\eta = 1$) | 18.00 | 51.93 | −33.93 |
| BDQN ($\eta = 2$) | 12.75 | 11.20 | 1.55 |
| A2C | 0.18 | 9.84 | −9.66 |

Table 3: Baseline bots and agent performance. Results are averaged over 5 seeds.

| Name | $R$ | Num. Episodes | Agg. Score |
|------|----:|--------------:|-----------:|
| DQN | 100 | 316300 | -653.59 |
| BDQN | 100 | 306770 | -15.50 |
| A2C | 100 | 161660 | -13.9 |
| DQN | 1000 | 65520 | -1143.5 |
| BDQN | 1000 | 75840 | 7.60 |
| A2C | 1000 | 17170 | -78.45 |

Table 4: Baseline bots and agent performance with longer recalls ($R \in \{100, 1000\}$). Results are averaged over 5 seeds.

## 4.2 Language Model Agent

Large language models (LLMs) have achieved state-of-the-art performance across a wide variety of natural language processing tasks. This is accomplished by simple token-level training objectives, applied to massive amounts of text data scraped from the web. LLMs can be further fine-tuned on specific tasks, and have been successfully utilized as components in game-playing systems, most notably Cicero which achieved human-level performance in Diplomacy (Meta et al., 2022). Even without fine-tuning, LLMs demonstrate some game-playing ability like finding legal chess moves, but exhibit poor performance at identifying checkmate-in-one moves (Srivastava et al., 2022).

Here we benchmark four model sizes (400M, 1B, 7B, 70B) from the Chinchilla family of LLMs (Hoffmann et al., 2022) on the RRPS task. As the information gain in RRPS per step is relatively low, to excel at this game, a successful agent has to anticipate the other agent's actions through the history. The language model agent is particularly relevant for RRPS because transformers have the unique ability to attend to (relative) patterns in the history that are important for prediction of the next token.

We utilize the LLM as a game-playing agent by selecting actions based on the model's prediction of what action the opponent will play next. The model is given a zero-shot prompt that plainly states the task and provides the game history (see Appendix A.3 for full prompt). The model's prediction of the opponent's next action is determined by choosing the max over the logprobs of the tokens `{R, P, S}`. The LLM agent then deterministically plays the action that beats the opponent's predicted action. The true actions played are appended to the prompt and the process is repeated. No parameters are fine-tuned at any point. Methodologies for prompting and fine-tuning LLMs and integrating them into larger systems are areas of

active research, and optimizing the LLM's performance on RRPS is beyond the scope of this paper. However, even in this simple zero-shot setting, and despite not having been trained on RRPS, LLMs demonstrate a surprising ability to to predict opponent actions that improves with model size. The largest LLM model achieves an average population return of 201.0 and aggregate score of 155.2, placing it fourth behind only GREENBERG, IOCAINEBOT, and BIOPIC; though, we did notice that size of the model size had a significant effect on the performance: our smallest model achieves an aggregate score of $-212.9$ in comparison (see Appendix A.3 for full results). Domain-specific fine-tuning would likely yield improvements and offers a promising direction for progress on this benchmark. Moreover, RRPS also offers a measure of an LLM's capacity for identifying and adapting to members of a population it interacts with.

### 4.3 Regularized Nash Dynamics

To minimize the exploitability (*i.e.* thus converging to a Nash equilibrium), a solution that empirically scale well is to learn a policy with the Regularized Nash Dynamics (R-NaD) algorithm (Perolat et al., 2022). In a nutshell, this method repeat a 3 step process: 1) building a reward transformation based on a regularisation policy, 2) a step where the process converges to a new fixed point of the game and 3) update the regularization policy with the fixed point found at step 2). With $R = 1$, R-NaD achieves **Pop. Return** in the set $[-10, -5]$, **W.P. Expl** in the set $[20, 40]$ and an **Agg. Score** in the set $[-50, -25]$ which is not far away from what the random policy achieves. The implementation used to produce these results uses the OpenSpiel implementation of R-NaD (Lanctot et al., 2019). We used the parameters from the open-source implementation and did a sweep over the following parameters (randomized over 5 seeds): $\eta$ reward transform : $[0.1, 0.2, 0.3, 0.4, 0.5, 0.6]$, trajectory max : $10, 000, 000$, batch size : $[64, 128, 256, 512]$, entropy schedule size : $(20000, )$, finetune from : $[-1, 300000, 600000]$.

This algorithm achieves a strategy that is hard to exploit but it will not exploit the other players.

### 4.4 Contextual Regret Minimization

RRPS fits into the setting of online learning and adversarial bandits, which looks at maximising value over repeated interactions in an environment with a fixed set of actions and unknown, dynamic payoffs. From the perspective of either player in a RRPS episode, they are repeatedly choosing an action of R, P, or S. The payoffs for each action are unknown because the player does not know their opponent's strategy for the next action. So, one natural choice for making decisions in RRPS is using an adversarial bandit algorithm. Regret minimizing algorithms all have theoretical guarantees that their average expected online performance is close to some optimal baseline, in hindsight. For example, an algorithm which minimises external regret is expected to do roughly as well any single static action $a$, if we looked back in time and asked how well we would have done if we had played $a$ instead. In RRPS, an agent that has low external regret would not have done significantly better by playing one the always-R, always-P, or always-S baseline policies against the opponent's sequence of actions. Other regret measures consider richer sets of baseline policies.

We look at four different algorithms for bandits with full information feedback, with different regret guarantees. Regret Matching (RM) is a simple, parameter-free algorithm which minimises external regret (Hart & Mas-Colell, 2000). Regret Matching+ (RM+) is a modification of RM that often has better empirical performance (Tammelin et al., 2015). RM+ also has a weak guarantee with respect to $k$-switching regret, which compares performance to all possible $k$-piecewise policies. The strongly adaptive online learner (SAOL) provides a strong guarantee for non-stationary environments, with a performance bound on any sub-interval (Daniely et al., 2015). SAOL is a meta-algorithm operating on top of another regret minimizing algorithm, and we used RM+ for the base algorithm in our implementation. Minimizing swap regret ensures that an agent would not have wanted to play action $a$ any time in the past when they had played $b$, for any actions $a$ and $b$. For swap regret, we used the meta-algorithm of Ito (Ito, 2020) on top of RM+.

While these four algorithms depend on the history – the historical actions played determine the current policy – they do not explicitly consider the current context ($R = 0$). One way to frame RRPS as a contextual regret minimization problem is to completely separate each possible recalled history for $R > 0$ into separate contexts, with independent regret minimizing algorithms running in each context. An agent using this discrete set of contexts has 9, 81, and 729 independent instances for $R = 1$, $R = 2$, and $R = 3$ respectively.

| Context | Agent | Pop. Return | W.P. Expl. | Agg. Score |
|---|---|---|---|---|
| $R = 0$ | RM | 48.45 | 27.39 | 21.06 |
| | SAOL | 67.30 | 34.73 | 32.57 |
| | RM+ | 59.66 | 26.36 | 33.30 |
| | SWAP-RM+ | 62.73 | 21.96 | 40.77 |
| $R = 1$ | SAOL | 178.08 | 91.32 | 86.76 |
| | RM | 164.75 | 76.30 | 88.44 |
| | RM+ | 169.88 | 63.99 | 105.89 |
| | SWAP-RM+ | 167.99 | 48.41 | 119.58 |
| $R = 2$ | SAOL | 171.46 | 155.39 | 16.07 |
| | RM | 175.43 | 148.89 | 26.54 |
| | RM+ | 174.44 | 121.79 | 52.65 |
| | SWAP-RM+ | 173.52 | 99.93 | 73.59 |
| $R = 1$ History Experts | RM | 157.55 | 23.92 | 133.62 |
| | RM+ | 157.99 | 17.51 | 140.48 |
| | SWAP-RM+ | 156.93 | 15.69 | 141.24 |
| | SAOL | 164.77 | 16.27 | 148.51 |

Table 5: Performance of regret minimizing agents across very historical contexts. Results are averaged over 5 seeds.

Another way to add context is to instead augment the environment actions with context experts that suggest environment actions: "history experts". For $R = 1$, we added six history experts suggesting the opponent's last action $o$, our last action $u$, the actions that beat $o$ and $u$, and the actions that lose to $o$ and $u$.

The full results from this experiment are included in Table 5. We see that, generally, SAOL performs best among all the choices of bandit rules, in all contexts. Overall, this context regret-minimizing agent is able to increase its population return as $R$ increases, but its exploitability also increases as well. In terms of aggregate score, the more dynamic definition of experts that are functions of the most recent actions $o$ and $u$ as well as their counter-strategies (history experts) perform significantly better than using the raw observation histories as context, which were also used by the baselines in Section 4.1. The best-performing version of this agent achieves an aggregate score of 148.51, which places it between third and fourth place among the bots in the population.

## 4.5 IMPALA and Generalization

In this subsection, we try a more modern implementation of a policy gradient algorithm that allows for bootstrapping and recurrent neural networks: Important-Weighted Actor-Learner Architectures (IMPALA) (Espeholt et al., 2018). IMPALA is a synchronous variant of (batched) A2C which uses importance-weighted corrections for its value function estimates, and has been show to work on visual environments such as the Atari suite (Bellemare et al., 2013) and at scale.

Specifically, we adapt the implementation provided in Haiku (Hennigan et al., 2020) to online (batched) agent consistent with the other agent implementations in OpenSpiel. We run two IMPALA agents against each other, similarly to the baselines in Section 4.1, sweeping over hyper-parameters policy learning weight $\in \{0.001, 0.0004, 0.0001\}$, entropy cost $\in \{0.01, 0.003, 0.001\}$, unroll length $\in \{20, 50, 100\}$, and $R \in \{1, 3, 5\}$. For IMPALA we use a basic recurrent network with two hidden layers of size $(256, 128)$ followed by an LSTM layer of size 256. After 600k episodes of training, the best population return and within-population exploitability achieved by this agent was 16.43 and 9.3, respectively (in both cases when $R = 1$) for an aggregate score of 7.13.

---

**Algorithm 1:** Population Reinforcement Learning (PopRL) for Two Players

---

**Input**: Bot population $\mathcal{B}$, batch size $B$, prediction weight $\rho$, probability of self-play $p$
**Input**: RL learning rule $\mathcal{L}$ (e.g. IMPALA)
**for** *batch number* $1, 2, \cdots$ **do**
    Reset data sets $\mathcal{D}_1 = \mathcal{D}_2 = \emptyset$
    **for** *episode* $t \in \{1, \cdots, B\}$ **do**
        **for** $i \in \{1, 2\}$ **do**
            Place PopRL agent $i$ in player slot $i$
            Sample $z \sim \text{UNIFORM}([0, 1])$
            **if** $z < p$ **then**
                Place PopRL agent $3 - i$ in player slot $j$
                Set opponent identification label $o$ to $|\mathcal{B}|$
            **else**
                Sample bot $b \sim \text{UNIFORM}(\mathcal{B})$ with index $b_{index}$, where $0 \leq b_{index} < |\mathcal{B}|$
                Set opponent identification label $o$ to $b_{index}$
            **end**
            Generate episode using agents $(i, 3 - i)$
            Add data from episode to $\mathcal{D}_i$ with combined loss: $\text{LOSS} = (1 - \rho)\text{LOSS}(\mathcal{L}) + \rho\text{LOSS}_{aux}$
        **end**
    **end**
    Perform separate learning steps on data sets $\mathcal{D}_1$, $\mathcal{D}_2$
**end**

---

### 4.5.1 IMPALA as a General Bot Exploiter Agent

Since IMPALA was designed to be a single-agent algorithm and was unable to significantly improve over the baseline algorithms, we now verify its ability to act as an approximate best response ("exploiter") agent when playing against the population. In this setup, a new opponent bot is uniformly sampled at the start of each episode to play against the IMPALA agent. By using similar hyper-parameter sweeps as before, we find a small set of good hyper-parameters (learning rate 0.0004, entropy cost 0.003, and vary only the unroll length $\in \{20, 50\}$). In this case, we find IMPALA can consistently reach a population return of 220 after 200k episodes, which is significantly higher than the independent RL setting.

One benefit of PBE is the ability to assess the capacity of an agent to generalize. In particular, we evaluate the ability of an IMPALA exploiter agent against bots that it has not trained to exploit. We apply cross validation over bot opponents: IMPALA trains against 33 agents, and evaluates only against the left-out set of 10 agents. We average the performance over 50 distinct sets of 10 left-out opponents. IMPALA consistently reaches an average of 120-130 per episode against the left-out bots, a significant drop compared to when training and testing opponent distribution are identical.

To investigate whether the generalization ability can be improved, inspired by UNREAL (Jaderberg et al., 2017), we augment the network and training procedure with an auxiliary task of opponent prediction. A new output head is added that predicts which specific opponent bot the agent is facing, and a standard classification loss is added to the combined RL loss with some prediction weight $\rho \in \{0.001, 0.01, 0.1, 0.5\}$. The results are shown in Figure 6 (in Appendix A.4). We observe that opponent identification helps, and improvements get are better with higher $\rho$. We also measure the average difference of the area-under-the-curve (interpreted as population return advantage per episode) between $\rho = 0.5$ and the baseline $\rho = 0$), achieving 12.94, 15.81, 13.00, and 11.05 at training episodes 25k, 50k, 100k, and 175k, respectively. The advantage diminishes slightly over time but maintains a significant positive advantage well into the training run.

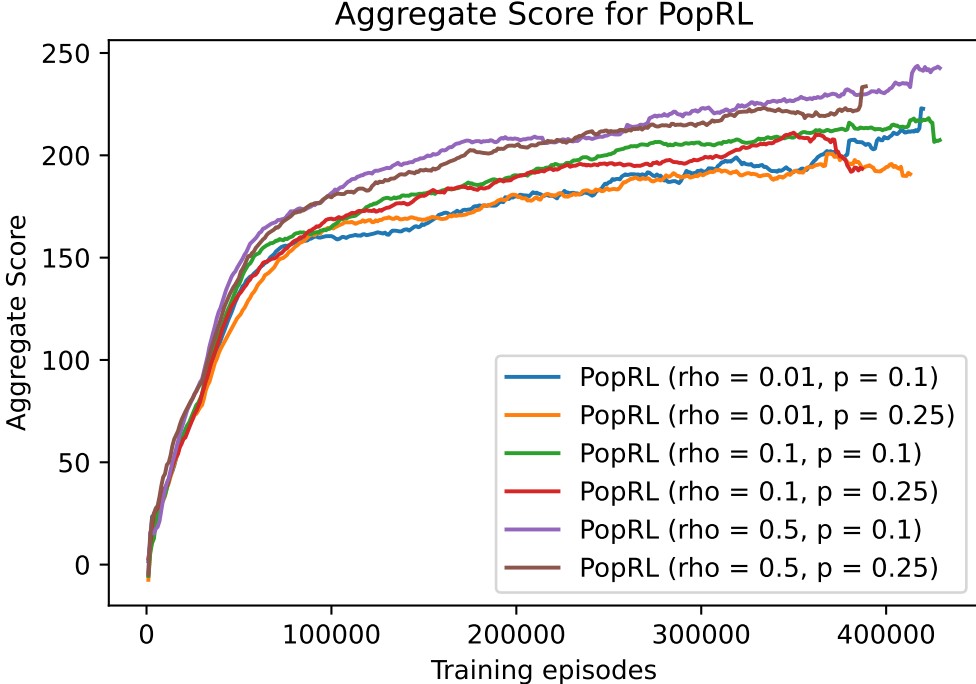

Figure 5: PopRL's aggregate score across hyperparameters. Each setting of hyperparameters was averaged across 5 seeds.

### 4.5.2    PopRL: A Hybrid Population-Based Training Algorithm

We now propose a new general training algorithm ("Population RL" in constrast to "IndRL") based on IMPALA with opponent identification. Inspired by Restricted Nash Response (Johanson et al., 2008) and game-theoretic population-based approaches (Lanctot et al., 2017; Hernandez, 2022; Strouse et al., 2021b), PopRL mixes between best responding to itself and to population members. Rather than train against the bot population only, a PopRL agent trains against an augmented population containing the 43 bots and an identical copy of another PopRL agent that is also independently training (concurrently or alternately). At the start of each episode, with probability $p$ the opponent is set to be the other PopRL agent, or (with probability $1 - p$) it is set to a uniformly sampled bot. In both cases, the agent uses opponent identification auxiliary task, but unlike before the number of classes is one greater to include identifying the other PopRL learning agent (44 instead of 43). The motivation is to leverage the population to train a generalist agent, while still guarding against being exploited by a similar learning agent. Pseudo-code is presented in Algorithm 1.

Results are shown in Figure 5. The best combination of hyper-parameters is able to achieve an aggregate score of 247.02, placing PopRL just behind IOCAINEBOT and far above BIOPIC, between second and third ranks. In addition, we show how the best PopRL agent scores against individual bots compared to GREENBERG in Figure 7 (Appendix A.4): while they score similarly on many of the agents in the population, they differ significantly against several bots.

We believe that PopRL combines several strengths in one approach: first, as was shown in Section 4.5.1, IMPALA equipped with a recurrent neural network and an opponent identification auxiliary task learns a very good bot exploitation strategy that generalizes across bots.

To ensure that it doesn't become exploitabile, another copy of the agent tries to find weaknesses ($100p\%$ of the time) while its learning its bot responses. In essence, this combination balances maximizing return and minimizing exploitability.

# 5 Discussion: Limitations, Related Work, and Future Directions

The purpose of this paper is to propose a new challenge for multiagent RL algorithms. While there has been impressive progress in MARL research producing agent that indicate human-level win rates in Go (Silver et al., 2016), Dota 2 (Berner et al., 2019), and Starcraft (Vinyals et al., 2019), humans (and in some cases, AI bots (Timbers et al., 2022; Wang et al., 2022)) could find counter-strategies that consistently exploit these agents. Our arguments in this paper are aligned with the ones in Player of Games (Schmid et al., 2021): that win rate alone is insufficient to determine human-level ability, and that game-theoretic reasoning is important to demonstrate robustness against exploits. However, unlike Player of Games which includes a complex search procedure, this domain focuses on repeated interactions with a population in the purely RL domain.

We highlight the property that RRPS is simple, has a low barrier to entry and yet complex dynamics when playing against a populations akin to Axelrod's tournaments in iterated prisoner's dilemma (Axelrod, 1984); additionally, RPPS allows the challenge to focus mainly on the tenuous balance between maximizing reward while not being exploitable. To the best of our knowledge, this a unique addition to the community: there is currently no challenge benchmark for learning agents that highlights these two axes in a single domain coupled with a population of human-crafted expert bots for finer-grained evaluation.

## 5.1 Limitations

The main limitation is that employing PBE requires a bot population and that the approximation quality of within-population-exploitability depends on there being bots that can exploit the various population mistakes within the repeated game. As such, the specific benchmark we are proposing and bot population we are characterizing are necessarily restricted to the domain of RRPS.

The population-based evaluation could be extended to other domains. To do so would require another set of hand-crafted bots. The aggregate score we are proposing in this paper only addresses the combination of maximizing reward and minimizing exploitability; a different domain may necessitate new metrics.

Finally, as mentioned above, this benchmark focuses on RRPS and the interactive between reward and exploitability. As such, we are not proposing PBE as an evaluation methodology for comparing multiagent RL algorithms more generally, such as in (Zawadzki et al., 2014). The values achieved by the learning agents we ran should be interpreted as yardsticks to challenge the community in a specific way, rather than providing an evaluation methodology for the broader problem of general MARL.

## 5.2 Related Work

RRPS has served as a dyadic test bed for understanding strategic interactions in both human and agents. The plurality of this research has focused on either learning opponent models or understanding learning dynamics. These two topics can be seen in the implementation of many of the competition bots—note, the majority of the related work occurs after the competitions. On opponent models, Cook et al. (2011) showed that imitation can be a powerful tool for learning strategies and understanding your opponent. Concurrent to this work, Lebiere & Anderson (2011) investigated human decision-making models, and notably discussed both that benefits of opponent models and their complications due to many models corresponding to the same behavior. Brockbank & Vul (2021) quantified human play by quantifying information gain regarding the behavior of your opponent.

The other plurality of work focuses on the evolving dynamics of players. Evolutionary algorithms have been the primary mechanism for understanding how learners develop over time (Ali et al., 2000; Bédard-Couture & Kharma, 2019). Wang et al. (2014) found that strategies often follow a cyclic pattern in the actions that are employed throughout a gameplay. However, studies have also discussed the possibility of chaos

accounting for the inability to learn in RRPS (Sato et al., 2002). Despite the apparent simplicity of RRPS, it is clear that there is still much we have to understand about learning dynamics within it.

### 5.3 Future Directions

There are several avenues of potential future work. Firstly, we one could try different populations in RRPS. Specifically, there are larger populations are openly available that could be easily adapted to fit within the PBE framework (Knoll et al., 2011).

Secondly, a more complex extension would be to introduce a continual version of RRPS with a dynamic population that can introduce or remove agents over time. This would test the general capability of an RL agent in a more dynamic way: could the agent also guard against anticipated new strategies, or learn now ways to counter these new strategies?

Finally, it could be interesting to see population-based evaluation methods applied to larger extensive-form games. There are several games being adopted by the community with hand-crafted strategies being used as a benchmark. Poker is a popular challenge domain; poker competitions were organized and run regularly as well as several man-machine poker competitions against human experts. Population-based evaluation could offer an alternative evaluation methodology, especially for games that current AI techniques have not mastered, such as variants beyond Texas Hold'em. There are several examples of domains in the cooperative AI (Dafoe et al., 2020), such as Hanabi (Bard et al., 2020; Hu et al., 2020) and Overcooked (Carroll et al., 2019; Strouse et al., 2021a), that might benefit from a replicable population-based evaluation rather than evaluation in self-play or one-time human/AI evaluations. And finally, population-based evaluation may be the only satisfying way to evaluate the quality of agents in general-sum or $n$-player games, such as Diplomacy, where there is no clear solution concept nor definition of "optimal strategy".

## 6 Conclusion and Future Extensions

We propose repeated Rock, Paper, Scissors, a population of previous tournament bots, and population-based evaluation as new challenge in sequential decision-making with multiple agents. The bots range widely in terms of population return, exploitability, and predictability. Several standard Deep RL baseline algorithms, that have attained human-level performance on various challenge domains, fail to achieve both high reward and to be robust to a population of RRPS bots.

We show that an LLM agent is able to achieve an aggregate score of 155.2, significantly higher than most baseline RL algorithms. The best agent trained via self-play (a contextual regret minimizer using SAOL) achieves an aggregate score of 148.51. When training against the population, IMPALA is able to to leverage opponent identification to learn general responses, and when combined with population-based training, achieves a high aggregate score of 247.02; but, even with the added information, it was unable to defeat the top two bots.

Several modern approaches are unable to defeat the top two hand-crafted bots, IOCANBOT and GREENBERG, that were submitted over 20 years ago. We invite the community to try their own approaches to achieve this result and contribute their findings in an effort to understand how learning agents could both maximize reward and reduce exploitability simultaneously.

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

# A  Additional Results

In this appendix, we give supplemental results referred to in the main text.

## A.1  Full Ranking of Bots

The performance and full ranking of bots in the population is given in Table 6.

## A.2  Hyperparameter Search

For Q-learning, we swept over learning rates $\alpha \in \{0.001, 0.02, 0.01\}$ and $R \in \{1, 3, 5, 10\}$. We observed that while different learning rates had differently-shaped curves, that ultimately the differences were small (with $\alpha = 0.02$ working best); on the other hand, the amount of recall made a significant difference.

For DQN and BDQN we swept over hyper-parameters batch size $\in \{32, 128\}$, $R \in \{1, 3, 5\}$, learning rate $\in \{0.02, 0.01, 0.001\}$, replay buffer capacity $\in \{10^5, 10^6\}$. For A2C we swept over hyper-parameters $R \in \{1, 3, 5\}$, $\lambda \in \{0.99, 0.9, 0.75\}$, entropy cost $\in \{0.01, 0.003, 0.001\}$, policy learning rate $\in \{0.0002, 0.0001, 0.00005\}$, critic learning rate $\in \{0.0001, 0.0002, 0.0005\}$. In all cases, networks were two-layer MLPs with layers of size $(256, 128)$ and ReLU activations except the final output layer.

## A.3  Language Model Agent

Language model prompt after two rounds of RRPS:

```
A repeated game of rock, paper, scissors is being played.
Guess the next move based on the game history.
Game history (player1, player2):
R,P
P,S
```

Minor variations in the prompt did not significantly impact performance. The scores are shown in Table 7.

## A.4  IMPALA Agent

## A.5  Behavioral Cloning

To access the extent to which the bots are predictable, we train action-prediction models that predict the bot's next action based on the full game history. We investigate three types of action-prediction models:

- *Individual*: a model trained to clone a single agent's behavior against the full population.

- *Population*: a model trained to the full population's behavior against the full population.

- *k-Fold*: a model trained to clone a fold ($n_{\text{in}}=30$) of the population, and is also evaluated for generalization on the held-out population ($n_{\text{out}} = 13$).

Hereafter, the sub-population being modelled is referred to as the *demonstrator* population/individual (e.g., in the case of the Individual model, it is the singleton bot). Common to all of the models is that the identity of the bots are never revealed.

Figure 4 shows results for the case in which a separate LSTM is trained per bot. In Figure 8 we compare average action prediction accuracy of individual LSTM models to a single LSTM model trained to predict next actions for a randomly sampled bot from the full population of 43 bots.

| Rank | Bot Name | Pop. Return | W.P. Expl. | Agg. Score |
|---:|---|---:|---:|---:|
| 1 | GREENBERG | 288.153 | 3.648 | 284.505 |
| 2 | IOCAINEBOT | 255.003 | 5.006 | 249.997 |
| 3 | BIOPIC | 196.365 | 36.665 | 159.700 |
| 4 | BOOM | 169.119 | 27.928 | 141.191 |
| 5 | SHOFAR | 152.008 | 16.865 | 135.143 |
| 6 | ROBERTOT | 177.767 | 50.154 | 127.613 |
| 7 | PHASENBOTT | 232.245 | 111.708 | 120.537 |
| 8 | MOD1BOT | 203.162 | 90.158 | 113.004 |
| 9 | SWEETROCK | 146.250 | 41.207 | 105.043 |
| 10 | PIEDRA | 146.080 | 41.441 | 104.639 |
| 11 | MARKOVBAILS | 111.192 | 17.601 | 93.591 |
| 12 | SUNNERVEBOT | 138.054 | 45.490 | 92.564 |
| 13 | MARKOV5 | 111.186 | 18.720 | 92.466 |
| 14 | ANTIROTNBOT | 121.387 | 58.616 | 62.771 |
| 15 | HALBOT | 212.429 | 176.229 | 36.200 |
| 16 | MIXED_STRATEGY | 114.131 | 83.488 | 30.643 |
| 17 | RANDBOT | 0.234 | 1.197 | −0.963 |
| 18 | PIBOT | 4.516 | 81.000 | −76.484 |
| 19 | ACTR_LAG2_DECAY | 146.319 | 236.865 | −90.546 |
| 20 | MARBLE | 148.661 | 240.988 | −92.327 |
| 21 | GRANITE | 149.252 | 241.840 | −92.588 |
| 22 | PREDBOT | 167.112 | 267.687 | −100.575 |
| 23 | ZQ_MOVE | 124.799 | 368.744 | −243.945 |
| 24 | MULTIBOT | 56.057 | 307.065 | −251.008 |
| 25 | TEXTBOT | −73.394 | 185.000 | −258.394 |
| 26 | DEBRUIJN81 | 10.250 | 301.679 | −291.429 |
| 27 | DRIFTBOT | −49.499 | 263.493 | −312.992 |
| 28 | ADDDRIFTBOT2 | −41.855 | 283.910 | −325.765 |
| 29 | RUSSROCKER4 | 172.334 | 529.751 | −357.417 |
| 30 | SWITCHALOT | −82.877 | 315.612 | −398.489 |
| 31 | ADDSHIFTBOT3 | −78.117 | 342.420 | −420.537 |
| 32 | FOXTROTBOT | −51.019 | 407.418 | −458.437 |
| 33 | FLATBOT3 | −71.952 | 416.524 | −488.476 |
| 34 | INOCENCIO | 17.616 | 579.868 | −562.252 |
| 35 | R226BOT | −212.619 | 399.845 | −612.464 |
| 36 | SUNCRAZYBOT | −83.609 | 578.089 | −661.698 |
| 37 | SWITCHBOT | −173.178 | 497.182 | −670.360 |
| 38 | PETERBOT | −174.238 | 927.986 | −1102.224 |
| 39 | FREQBOT2 | −341.744 | 999.000 | −1340.744 |
| 40 | COPYBOT | −475.327 | 997.000 | −1472.327 |
| 41 | ROTATEBOT | −602.641 | 998.121 | −1600.762 |
| 42 | ROCKBOT | −610.116 | 1000.000 | −1610.116 |
| 43 | ANTIFLATBOT | −648.420 | 999.002 | −1647.422 |

Table 6: The full ranking of bots in the population.

| Bot Name | Chinchilla Parameters Size | | | |
| --- | --- | --- | --- | --- |
| | 400M | 1B | 7B | 70B |
| AC_L2_DECAY | −123.3 | −1.5 | −28.0 | −13.1 |
| ADDDRIFTBOT2 | 27.5 | 52.4 | 82.4 | 89.7 |
| ADDSHIFTBOT3 | 73.5 | 188.0 | 222.6 | 155.5 |
| ANTIFLATBOT | 995.0 | 995.6 | 991.4 | 992.6 |
| ANTIROTNBOT | 51.4 | 55.8 | 59.4 | 60.9 |
| BIOPIC | −193.5 | −63.4 | −52.8 | −20.0 |
| BOOM | −65.7 | 10.4 | −6.8 | 9.5 |
| COPYBOT | 981.0 | 981.0 | 983.0 | 979.0 |
| DEBRUIJN81 | −51.0 | −11.0 | −30.0 | −20.0 |
| DRIFTBOT | 80.3 | 123.4 | 182.6 | 155.4 |
| FLATBOT3 | 106.6 | 148.2 | 106.5 | 154.2 |
| FOXTROTBOT | −1.7 | 57.3 | 44.8 | 33.9 |
| FREQBOT2 | 598.0 | 774.0 | 871.0 | 919.0 |
| GRANITE | −16.8 | 120.6 | 156.8 | 128.6 |
| GREENBERG | −305.9 | −121.2 | −108.6 | −39.8 |
| HALBOT | −300.9 | −145.6 | −134.9 | −8.9 |
| INOCENCIO | 449.3 | 337.6 | 793.1 | 382.1 |
| IOCAINEBOT | −323.0 | −144.4 | −148.7 | −28.6 |
| MARBLE | 20.6 | 141.7 | 146.1 | 123.0 |
| MARKOV5 | −78.6 | 3.5 | −14.4 | −19.3 |
| MARKOVBAILS | −80.4 | 2.4 | −10.9 | −21.1 |
| MIXED_STRAT | −15.4 | 31.2 | 34.0 | 57.4 |
| MOD1BOT | −206.9 | −87.7 | −76.2 | −25.0 |
| MULTIBOT | 198.0 | 211.0 | 366.0 | 224.0 |
| PETERBOT | 652.1 | 815.2 | 831.0 | 846.2 |
| PHASENBOTT | −315.3 | −174.7 | −165.4 | −45.8 |
| PIBOT | −2.0 | −11.0 | 1.0 | 9.0 |
| PIEDRA | 42.4 | 42.8 | 44.4 | 44.7 |
| PREDBOT | −143.2 | 12.3 | 24.4 | 67.6 |
| R226BOT | 372.4 | 364.3 | 370.8 | 344.4 |
| RANDBOT | 1.1 | 3.4 | 6.3 | −3.7 |
| ROBERTOT | −94.5 | −8.6 | 3.0 | −5.8 |
| ROCKBOT | 998.0 | 998.0 | 996.0 | 994.0 |
| ROTATEBOT | 983.0 | 992.0 | 995.0 | 1000.0 |
| RUSSROCKER4 | −234.4 | −55.1 | −55.8 | −14.5 |
| SHOFAR | −80.2 | −34.5 | −23.6 | −15.8 |
| SUNCRAZYBOT | 292.5 | 389.7 | 423.2 | 466.3 |
| SUNNERVEBOT | −141.3 | −45.7 | −37.0 | −16.5 |
| SWEETROCK | 43.9 | 49.6 | 30.8 | 45.3 |
| SWITCHALOT | 116.5 | 123.9 | 115.1 | 154.5 |
| SWITCHBOT | 200.1 | 230.7 | 225.1 | 276.6 |
| TEXTBOT | 144.0 | 113.0 | 129.0 | 31.0 |
| ZQ_MOVE | 80.4 | 154.9 | 196.2 | 196.1 |
| POP. RETURN | 110.1 | 177.2 | 198.6 | 201.0 |
| W.P. EXPL. | 323.0 | 174.7 | 165.4 | 45.8 |
| AGG SCORE | −212.9 | 2.5 | 33.2 | 155.2 |

Table 7: LLM agent performance against bot population (avg over 10 runs).

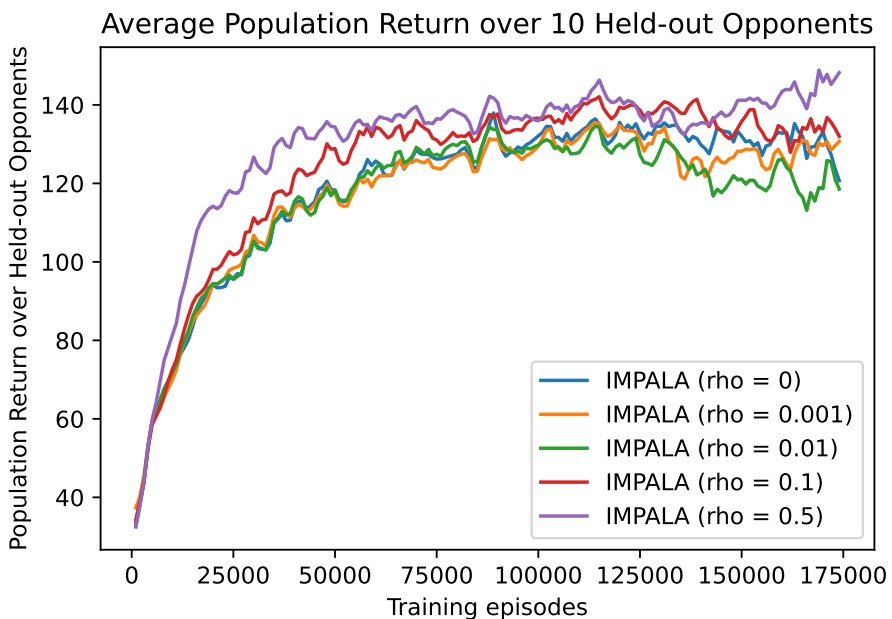

Figure 6: Population return over held-out opponents when IMPALA is trained as an exploiter agent. Each method is averaged over 50 held-out splits.

**Training**   The models are trained with a behavioral cloning objective that maximize the action-prediction model's likelihood of playing a demonstration action (from the bot). Demonstration data is generated dynamically by uniformly sampling a demonstrator and co-player. Note, that the co-player is sampled uniformly from the full population for bot the Individual and Population models, but is sampled only from the within-fold population for the $k$-Fold model. Data is generated in parallel by 20 processes populating a temporary data buffer that is uniformly sampled to prevent correlation in complete batches from the same strategy profile. The training batches contain 128 sub-trajectories of length 20 providing a limited recall during training, but during evaluation full recall can be maintained within the learned memory. Each model is trained for 1B frames corresponding to 1M episodes.

**Evaluation**   The trained models are fixed and their predictability is measured by their agreement with a demonstrator playing 100 episode for each unique profile (across both demonstration- and co-player-bots). Agreement is measured by average action accuracy across all episodes.

**Model Implementation**   The models are implemented with a 2-layer LSTM with sizes $[64, 64]$. The output of final layer of the LSTM is projected into action space by an 3-layer fully-connected neural network with sizes $[64, 32, 3]$.

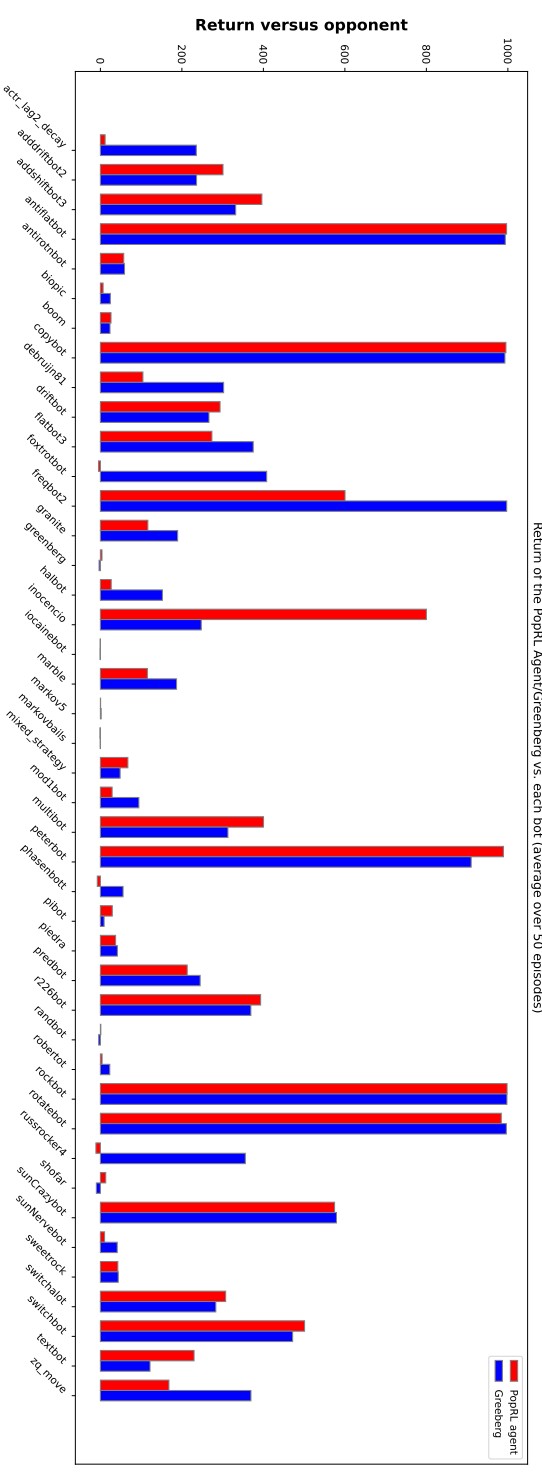

Figure 7: Population return of PopRL agent against individual bots compared to Greenberg.

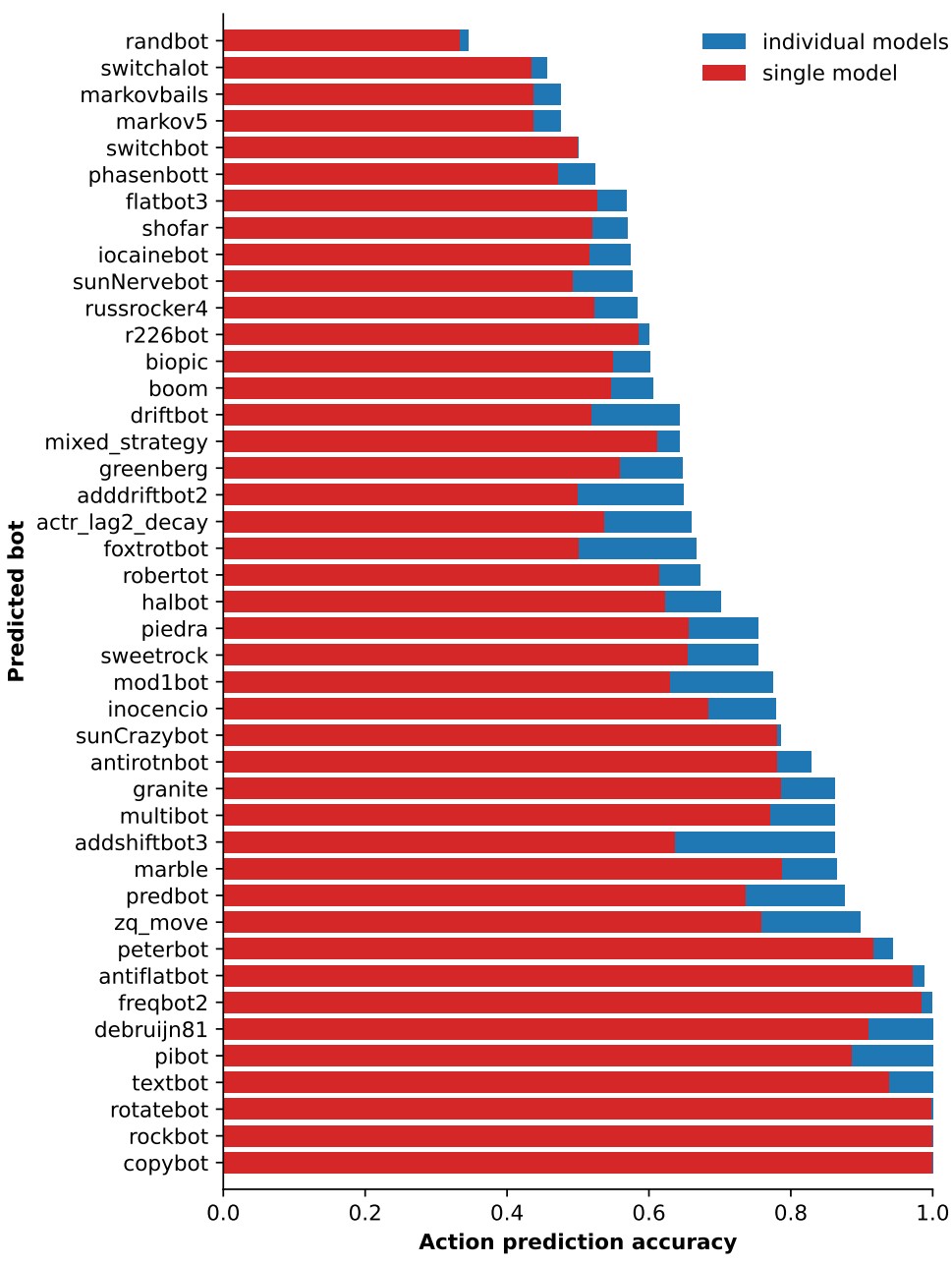

Figure 8: Average action prediction accuracy comparison between individual LSTM models and a single LSTM model cloning all bots.

