# OpenReview forum: "Population-based Evaluation in Repeated Rock-Paper-Scissors as a Benchmark for Multiagent Reinforcement Learning"
_TMLR — Accepted by TMLR_

### Review · Reviewer_wm2S · 2023-07-09

**Summary Of Contributions:**

This work argues that repeated rock-paper-scissors (RRPS), a well studied game in the game theoretical literature, can be used as a benchmark for multi-agent reinforcement learning. The authors primarily aim to provide a system for studying the problem of balancing average return versus best-case robustness against a worst-case opponent.

To build the environment, they employ bots previously made for other RRPS competitions, forming a population of bot players that agents can be compared against.

The manuscript focuses on analyzing the properties of this population of agents, as well as comparing the performance (wrt. the benchmark) of a few common RL baselines.


**Audience:**

Yes

**Claims And Evidence:**

Yes

**Requested Changes:**


- A better justification of the importance of the problem for folks working outside of game theory would be great to have. What are the (positive) consequences of finding learning algorithms that can effectively trade worst-case performance vs average returns in these settings? I would also be careful with over-claiming here, as noted in the above section.

- I don't understand why the benchmark is usually testing on a sample of the bots rather than the entirety of the population. Isn't it kind of the point that one might wish to understand the distribution of performance against qualitatively different opponents? In that case, isn't it necessary for such a low number of opponents to test against all of them? Some clarity around the reason behind these choices would be good.

- Tighten the story behind the baselines. What are the baselines trying to tell the reader? How far have they pushed the benchmark? (Suspicion: does the benchmark becomes "solved" if deep policies are given a much longer history window?)

- It is IMO unacceptable to write something like this: "The precise LLM used and number of parameters will be revealed in the camera-ready copy, if accepted for publication." -- the manuscript is already public, so why the secrecy? The lack of such information makes Section 4.2 largely pointless for the reader. Please add this information (as well as prompt employed, and other such relevant information to make it possible to replicate the result).

### Typos

- Page 2
  + Identical (Note: they hopefully aren't?)
  + Would a table
  + Unless [...], use R = 1...


**Strengths And Weaknesses:**

## Strengths

- The manuscript presents a straightforward idea executed in a good idea. The benchmark seems to be well packaged, and the paper presents a reasonable amount of information to the hypothetical reader looking to employ it in their research.

- Particularly solid is the usage (and description) of  each agent that makes the benchmark, whose simplicity yet great emerging dynamics are definitely a big strength (since it gives the benchmark a lot of degrees of freedom in terms of evaluating future work).

## Weaknesses

- I found the problem that this benchmark is trying to solve -- i.e balancing (a) average return versus (b) best-case robustness against a worst-case opponent -- to be poorly justified. It is definitely true that MARL has an evaluation problem that single-agent RL does not, but IMO that stems primarily from MARL being not a single problem definition, but instead a (large) categories of problems. The manuscript presents regularizing (a) vs (b) as if a good combination of the two would be sufficient to solve _a_ general MARL problem, but there's a large set of issues that remain (say, e.g. zero-shot / one-shot / ad hoc multi-agent stuff). Given that (b) seems to stem primarily from game theory, I'd perhaps expect the manuscript to draw a better justification from that large literature of why this is a problem worth caring about (because it's not clear to me in general practical terms).

- With all due respect, the baselines story (i.e. Ch. 4) from my perspective could use some work, and the comparisons against the bots feels particularly a bit unfair.
  + Section 4.1 presents agents whose observability wrt. history is limited to max 10 joint-action steps, which seems a weird limitation given that (a) the state space is so simple, (b) the game dynamics are trivial but intrinsically stochastic, and (c) the amount of information gained per step is relatively low. "Recall" is bound to be a bottleneck against bots that can do long history matching, and seems like something that could be dealt with fairly trivially with some simple state abstraction or representation learning.
  + It feels like a couple of the subsections were put there just to (superficially) cover relevant topics. Section 4.2 is very light on details, and doesn't seem to be that relevant (why should it matter *for this benchmark* that LLMs can play RRPS?). Section 4.4 also has no explained relevance to the benchmark (which, to be clear, I'm sure it obviously does -- but it's just not in the manuscript).

- Given the wide range of bots available in the dataset, I was expecting the manuscript to use baselines to provide a large degree of insight about the kinds of bot-agent games that arise from using these benchmark. However, the evaluation often stopped at the level of average performance wrt. the metrics that the authors cared about. Seems like a missed opportunity to kickstart research on the benchmark appropriately.

EDIT: cleaned up my wording on Section 4.1 -- apologies for the miscommunication, I didn't mean for the comment to be dismissive of the work.

---

### Review · Reviewer_RrDa · 2023-07-10

**Summary Of Contributions:**

The paper proposes a benchmark for multiagent learning that evaluates the agents on the repeated rock-paper-scissors (RRPS) game, against 43 bots submitted during two RRPS competitions held in the early 2000s. Three metrics are proposed and various well-known learning algorithms are evaluated. The paper also proposes a new training protocol, dubbed PopRL, that bases on IMPALA, UNREAL, and Restricted Nash Response.

**Audience:**

Yes

**Broader Impact Concerns:**

N/A.

**Claims And Evidence:**

No

**Requested Changes:**

See the "Strengths And Weaknesses" section.

**Strengths And Weaknesses:**

Strengths:
* Challenging benchmarks for learning are important and has historically driven research in many areas of machine learning. Consequently, contribution in this area in the domain of multiagent learning is a promising endeavor.
* The paper evaluates a range of well-known algorithms and uses a couple of known libraries, including Openspiel.
* A new algorithm, PopRL, is proposed.
* The paper asks some interesting questions: Is it possible to train a model (here LSTM) to predict bots' actions? Is it possible to reason about the whole population of bots by learning on replays from games played between agents in a subpopulation? Can RL agent learn to achieve good results in the population and not be exploitable? What are the capabilities of LLMs in RRMS or identifying the opponent? Can IMPALA agent perform good as an independent agent or as an exploiter, or do the auxiliary tasks help?

The paper lacks in the following aspects:
* Focus:
	* What are clear objective of the benchmark and why is it relevant? What exact deficiencies of the current learning protocols can a RRPS highlight? Can the benchmark be used to reason about the algorithms weaknesses and point out towards future research directions.
* Clarity of exposition:
	* A big concern is that, although the paper provides various experiments, it only marginally discusses the results.
	* What one would typically expect from a benchmark paper is an analysis of existing algorithms' behavior on the benchmarks, highlighting the limitations, and hypothesizing about underlying causes.
	* The reader would also expect from a paper like this, some guidance concerning future research or a list of open problems.
 * Structure:
	 * There is no related work section (information in this respect is scattered across the text, most notably Section 4). Including such a section would help to organize the paper better, keep the experimental part cleaner and focused on discussion of the results, reassure the readers that the Authors did a proper literature overview, and help with future research.
	 * There is no limitations section. Such a section is always useful: it not only shows that the Authors are aware of various constraints associated with the work, but allows researchers to avoid pitfalls and speed-up research.
	 * Technical discussion not directly relevant for the text (such as the hyperparameters grid search) can be moved to Appendix.
	 * The paper would benefit from one central piece of information (e.g., a Table) where all the results of all the tested methods are summarized.

Other:
*  The paper could use some of the guidelines for RL evaluation from [1] (as to the number of seeds or uncertainty bounds).
* Table 3 seems to have an error: the three metric values in row QL (R=10) do not reconcile. Also a comment regarding this number should be reviewed (last sentence in Section 4.1)
* The last sentence on page 7 is not clear ("averaged over a sliding window of 50 most recent evaluations ")
* On page 4, it is written "Figure 2 below", where in fact the Fig 2 is above. Additionally, the a consistent naming should be used (see e.g., "RoShamBo" in caption of Figure 2).
* The choice for the aggregated metric could be discussed more (e.g., that it has some desirable properties).
* The paper only allocates one fourth of a page to discuss and describe a new algorithm it proposes.
* In footnote 3, it is written "The precise LLM used and number of parameters will be revealed in the camera-ready copy, if accepted for publication." The paper should contain all the necessary information to make its assessment possible.

To summarize, after a revision suggested above, the paper could be accepted, as some of the TMLR's audience would most likely be interested in its contents.

[1] Agrawal, R., et al. Deep Reinforcement Learning at the Edge of the Statistical Precipice.

---

### Review · Reviewer_KsZo · 2023-07-11

**Summary Of Contributions:**

The paper proposes the repeated rock-paper-scissors game as a benchmark for multi-agent learning. More specifically, the authors have collected $43$ bots of various strengths against which a new agent might be compared. Two principal metrics, which are used, are $\texttt{PopulationReturn}$ and $WithinPopExpl$. The former measures how an agent performs 'on average', while the latter captures the worst-case scenario, i.e., how much it can be exploited by some other agent. $\texttt{AggregateScore}$ is used to aggregate these two into one score.

**Audience:**

Yes

**Claims And Evidence:**

Yes

**Requested Changes:**

See above.

**Strengths And Weaknesses:**

The paper is very clear and precise in presentation, with smooth execution. I enjoyed the discussion as well as the number of analyses and the discussion overall.

I have a few critical comments as well.

1. Sec 4.1 is probably not very informative. I expect (please correct me if I am wrong) that the training algorithm, which does not maintain a population of agents, necessarily falls into 'local minima' strategies and, as such, cannot reliably provide much information. I'd probably keep it in the appendix only.
2. The results in Fig 5 seem to be growing. I'd appreciate it if at least one run is taken to the 'asymptotic performance.'
3. The results in Fig 5 how many seeds they are using. If only one (as nothing else is said), then the conclusions in this section are quite doubtful on the statistical end.
4. It is somewhat unsatisfactory that only one network size is studied for the RL setting. It is quite

Apart from these specific comments above, I am unsure if the rock-paper-scissor game is challenging enough to be a meaningful benchmark for some not-too-short time.

---

### Decision · Action_Editors · 2023-09-18

**Recommendation:** Accept with minor revision

**Comment:**

The paper proposes a benchmark for multiagent reinforcement learning based on the repeated rock-paper-scissor game.
The reviewers found the paper well-written, addressing an interesting topic, and providing a solid benchmark.
On the other hand, they raised several concerns (lack of motivation and focus for the proposed benchmark, poor discussion of the results, related works, and limitations).
The authors addressed most of these concerns and uploaded a new, improved version of their paper.
However, not all the changes proposed in their rebuttals have been implemented (e.g., "Figure 2 below" and the values in Table 3).
To accept the paper, the authors have to address all the points discussed in their rebuttals.

**Audience:**

The paper is of interest to a part of TMLR's audience.

**Claims And Evidence:**

The paper supports its claims with empirical evidence.